# Importance of Anatomical Variation of the Hepatic Artery for Complicated Liver and Pancreatic Surgeries: A Review Emphasizing Origin and Branching

**DOI:** 10.3390/diagnostics13071233

**Published:** 2023-03-24

**Authors:** Kapil Kumar Malviya, Ashish Verma

**Affiliations:** 1Department of Anatomy, Institute of Medical Science, Banaras Hindu University, Varanasi 221005, Uttar Pradesh, India; 2Department of Radiodiagnosis and Imaging, Institute of Medical Science, Banaras Hindu University, Varanasi 221005, Uttar Pradesh, India

**Keywords:** hepatic artery, MDCT, pancreatoduodenectomy, liver transplantation, aberrant artery

## Abstract

Knowledge of anatomical variations of the hepatic artery from its origin to intrahepatic segmentation is of utmost importance for planning upper abdominal surgeries including liver transplantation, pancreatoduodenectomy, and biliary reconstruction. The origin and branching pattern of the hepatic artery was thoroughly described by the classification of Michels and Hiatt. Some rare variations of the hepatic artery were classified by Kobayashi and Koops. By the use of the multidetector computed tomography (MDCT) technique, the branching pattern of the hepatic artery can be visualized quite accurately. Unawareness of these arterial variations may lead to intraoperative injuries such as necrosis, abscess, and failure of the liver and pancreas. The origin and course of the aberrant hepatic arteries are crucial in the surgical planning of carcinoma of the head of the pancreas and hepatobiliary surgeries. In liver transplant surgeries, to minimize intraoperative bleeding complications and postoperative thrombosis, exact anatomy of the branching of the hepatic artery, its variations and intrahepatic course is of utmost importance. This review discusses variations in the anatomy of the hepatic artery from its origin to branching by the use of advanced imaging techniques and its effect on the liver, pancreatic, biliary and gastric surgeries.

## 1. Introduction

Hepatic vasculature is nourished by dual circulation in the form of the portal vein and hepatic artery. Hepatic arterial vasculature has a central role in hepatobiliary perfusion. Normally common hepatic artery (CHA) is originated from the celiac trunk. During the course, CHA bifurcates into proper hepatic and gastroduodenal arteries. The proper hepatic artery enters the parenchyma of the liver through hepatic hilum and divides into the left (LHA) and right hepatic arteries (RHA). During the course, hepatic artery comes in relation to the upper margin of the body of the pancreas and peritoneum of the posterior abdominal wall [1]. To know the exact location of hepatic artery variations, their 3D anatomical relation and course, advanced radiological imaging such as multidetector computed tomography (MDCT) scans and magnetic resonance imaging (MRI) are very helpful and play a key role. According to the literature, hepatic artery variations are common, and knowledge of these variations has great clinical significance in complicated liver and pancreatic surgeries. Lack of knowledge of these anatomical variations may result in intraoperative injuries in open and laparoscopic surgical procedures [2]. Hepatic artery variations were studied in detail by various researchers and given the internationally accepted classification. The classification given by Michel and Hiatt is the most popular and acceptable [3,4,5]. Michel classified hepatic artery variations into 10 different types and Hiatt classified them into six different types [4,5] (Table 1). According to both, Type I variation is the most common and represents regular branching of the hepatic artery [4,5,6]. Type III variation is the most common abnormal variation affecting upper abdominal surgeries [7,8,9]. In the Type III variation, RHA is entirely replaced or aberrant and initiates from the superior mesenteric artery (SMA). The aberrant hepatic artery needs special attention during open and laparoscopic pancreatic surgery as RHA may pass adjacent, infiltrates the head or uncinate process of the pancreas prone to injury during surgery and run-in relation to the common bile duct, finally arriving at the hepatic hilum. According to Mangieri et al., the presence of an aberrant RHA (aRHA) itself acts as an aggravating factor for the development of hepatic metastasis disease [10]. Similarly, Michel Type IX variation (Hiatt Type V), in which SMA give rise to CHA is highly prone to injury during the resection of the uncinate process during the time of pancreatic surgeries [4,5,11]. The common bile duct receives arterial supply from RHA and any intraoperative damage to RHA during hepatobiliary surgeries leads to hepatic and biliary ischemia, resulting in abnormal liver functioning, anastomosis leak, and possible liver failure [11,12]. Thus, accurate knowledge of these variations is very important for the preoperative preparation of pancreatic and hepatobiliary surgeries to minimize intraoperative injuries and complications. Along with the study of variations of origin of the hepatic artery, knowledge of intrahepatic branching and intersegmental communication of the hepatic artery is also very important and can be visualized accurately with the help of an MDCT angiogram. The importance of intersegmental arterial communication between the segments of the liver comes into the picture during the segmental liver resection or during orthotopic liver transplantation for the maintenance of proper liver collateral circulation and prevention of liver ischemia.

The present review of the literature emphasizes upon different anatomical variations of origin and branching of hepatic arteries and their clinical importance in complicated hepatic and pancreatic surgeries such as liver transplantation, pancreatoduodenectomy and biliary reconstruction. The exact knowledge of 3D anatomy of hepatic artery variations is possible by the use of new advanced imaging techniques such as MDCT and is very useful in improving the surgical outcome in upper abdominal surgeries.

## 2. Embryology

For understanding different types of variations of the hepatic artery, it is important to understand its embryology. Embryologically, vascular supply of the hepatobiliary system and gut comes from four interconnecting ventral roots known as ventral segmental arteries originating from the dorsal aorta. These four roots give rise to the left gastric, splenic, common hepatic and superior mesenteric arteries, respectively [2,13]. The development of the hepatic artery starts in the 8th week of intrauterine life [14]. The intrahepatic branches of the embryonic hepatic artery are first seen around the 10th week of intrauterine life in the central zone of the fetal liver along with the intrahepatic branches of the embryonic portal vein [14]. The developing portal venous system of the liver acts as a framework for the development of the hepatic arterial system. The hepatic arterial system develops in close relation and coordination with the developing hepatobiliary and portal venous systems [15]. According to the available data, hepatic arterial vasculature does not form from a single artery. There are three different parts in the development of liver vasculature including the left lateral part, right lateral part and middle part. The left lateral part is represented by the left hepatic branch and gives rise to the left gastric artery (LGA). The right lateral part is represented by the right hepatic branch arising from the omphalomesenteric artery. The middle hepatic artery comes from the junction of the common hepatic and gastroduodenal artery also called the proper hepatic artery representing the middle branch. Normally during the course of the development, the left and right hepatic arteries disappear, and only the middle hepatic artery persists as the main blood supply of the liver [16]. Non-disappearance of the left and right hepatic arteries or persistence of their interconnection may give rise to aberrant or various types of arterial variations [17].

## 3. Hepatic Artery Variations on the Basis of Anatomical and Radiological Aspect

Variations of the hepatic artery anatomy are most prevalent and seen in more than half of the population. A wide variation is seen in the origin and branching of the hepatic artery, which is detected by cadaveric dissection or radiologically in the living person by the use of MDCT [4,5,18,19]. According to Michel, in 55% of cases, the hepatic artery originated from the celiac trunk, but some researchers noted normal hepatic artery anatomy in 70–80% of cases [20,21] (Figure 1). In some cases, apart from the regular arterial supply, the liver also receives the extra branch originating from the left gastric or superior mesenteric artery identified as the accessory branch. In another condition, normal left and right hepatic arteries are absent, and arterial supply of the liver comes from other sources termed the replaced hepatic artery [4].

### 3.1. Variations of Common Hepatic Artery

The variation of CHA ranges from 0.5 to 3.5% [4,5,19]. Two types of variations are most commonly observed in CHA, as it arises directly from the aorta (Hiatt Type VI) (Figure 2) or branch of the superior mesenteric artery (Michel Type IX or Hiatt Type V) forming the hepatomesenteric trunk [4,5] (Figure 3). Both variations of CHA have great significance on the surgical outcomes in hepatobiliary and pancreatic surgeries. When CHA arises from the aorta, it follows the normal course to reach the liver hilum. At the liver hilum, the proper hepatic artery (PHA) divides into RHA and LHA. RHA is longer and has an independent segment V branch proximally. A segment VI branch is formed as the artery continues postero-superiorly, and subsequently, segment VII and segment VIII branches are formed. LHA in the left lobe divides into segmental branches and supplies segments II and III [22]. Segment IV has a peculiarity in its blood supply as it is supplied by a branch from LHA or RHA or sometimes a hilar branch originating from CHA or PHA at the junction of origin of RHA and LHA, which is termed as the middle hepatic artery (MHA) [23]. Segment I also shows much variability in arterial supply, as in the maximum number of cases, it is supplied by the branches from both RHA and LHA followed by the branches from RHA or LHA separately [22]. These segmental branches also communicate with each other and form intersegmental arterial communications. Various studies describe the intralobular and intersegmental arterial communicating arcades of liver through anatomical dissection and radiological imaging [24,25,26]. At the hilar plate of the liver, the communicating arcade is present between the right and left hepatic artery in the form of the hilar plexus. It plays an important role in intralobular arterial communication and also supplies the hilar biliary tract [27]. The communicating arcades are present proximal to the portal vein bifurcation and extrahepatic at the hepatic hilum. The caudate lobe of the liver receives the arterial supply from segment I artery as well as from communicating arcades [26]. Some studies describe the intersegmental arterial communication between the medial and left lateral segments of the liver. These intersegmental arterial communications of the liver are maintained by the medial and left lateral hepatic arteries. The plexus around the hilar plate of the liver has an important role in maintaining the collateral circulation between the segments of the liver [28].

The variations of CHA need great attention at the time of surgery, as injury to the hepatic artery not only causes liver ischemia but may also cause huge intraperitoneal life-threatening blood loss [2]. In the second variation, CHA originates from SMA as an aberrant artery and further follows two paths (intra- and extra-parenchymal paths) to reach the liver hilum and follow the conventional intrahepatic course to supply the segments of liver. In the extra parenchymal path of CHA, after origin from SMA, it reaches the liver by passing outside the pancreas in relation to the posterior surface of the pancreatic head and portal vein. In the intra-parenchymal path, CHA after originating from SMA passes through the pancreatic head to reach the liver and is known as transpancreatic CHA. It is very difficult to save CHA in case of its intraparenchymal course during the surgical dissection of the pancreatic head. In the condition of damage to the CHA, reconstruction surgery is performed with end-to-end anastomosis [2,19]. In cases where hepatic arterial reconstruction cannot be performed, the integrity of the gastric arterial arcade between the right and left gastric arteries is detected to maintain the hepatic perfusion via collateral circulation. In case of resection of CHA, preservation and detection of the gastric arterial arcade is necessary to maintain the perfusion of liver, to maintain the flow of the right posterior hepatic artery through routes of the celiac trunk via gastric arterial arcade from LGA to right gastric artery (RGA) [29]. Such useful variations which may be used during salvage procedure are also well seen in the MDCT angiogram.

### 3.2. Variations of Right Hepatic Artery

RHA variations are commonly observed. aRHA is the most commonly present variation of the hepatic artery in the population with an incident rate of 15 to 35% [4,5]. According to some literature, the incidence rate is as high as 49% [30,31]. aRHA is one of the most clinically prevalent conditions to be attended carefully in patients undergoing pancreaticoduodenectomy. It may originate from the SMA, proper hepatic artery, aorta, left gastric artery and splenic artery. According to the present literature, replaced RHA is present in 5 to 21% of cases and the accessory RHA is present in 1 to 8% of cases [2,4,5,18,19,30,32]. Replace RHA (Michel Type III) arises from the SMA, passes laterally and posterior to the portal vein and posterolateral to the bile duct, enters the hepatoduodenal ligament and can be palpated at the foramen of Winslow [2,4,18] (Figure 4). During the course, replace RHA passes behind or through the head of the pancreas or close to the hepatocystic triangle, making it more susceptible during laparoscopic pancreatoduodenectomy and cholecystectomy [30]. In 2% of cases, aRHA arises from the aorta; very rarely, less than 1% of cases originated from the splenic and left gastric artery. According to some researchers, aRHA originates from the celiac trunk and CHA passes superior to the common bile duct identified as the superior aRHA, and that which arises from the SMA passes inferior to the common bile duct known as inferior aRHA. The later needed more clinical attention at the time of pancreatic and biliary surgery [2,30].

The diameter of aRHA is also an important beneficial factor in some clinical conditions of the liver. aRHA originates from the major vessel, which has a larger diameter and also supplies a larger area of liver lobes. The replaced RHA has a beneficial effect in liver transplantation as it supplies a larger area of the liver and provides a larger artery for the anastomosis [30]. Accessory RHA supplying the liver has a beneficial effect in the form of collateral circulation to prevent liver ischemia in case of main vessel injury [2,19]. In the Michel type III variation, intrahepatic branching of the hepatic artery shows mainly three types of patterns as the IV segment of liver is supplied by the branch from RHA, LHA or branches from both RHA and LHA as dual blood supply [3,23,33]. The dual blood supply of segment IV has the clinical advantage at the time of liver transplantation surgeries as in the case of obstruction of one artery; collateral circulation is maintained by the opposite blood vessel. The collateral circulation of the liver also plays an important role at the time of ligation of RHA in the situation of trauma or elective surgery in case of aRHA, which is the replacement that originated from the SMA and passes behind the common bile duct, rarely resulting in the liver ischemia because the hepatic arterial arcade of the left hemiliver and intersegmental collaterals maintains the blood supply of the right hemiliver through communicating branches or hilar marginal artery [34,35].

### 3.3. Variations of Left Hepatic Artery

LHA variations are the second most common among hepatic artery variations and range from 10 to 30% of the total variations. Aberrant LHA originates from the left gastric artery, celiac trunk, aorta, CHA, RHA, gastroepiploic artery and very rarely from SMA and splenic artery [36]. According to the available literature, most commonly, aLHA originates as a branch of the left gastric artery observed in 5–10% of the population. Replace LHA (Michel and Hiatt Type II variation) observed in 2–10% of the population (Figure 5) and accessory LHA (Michel Type V and Hiatt Type II) are seen in 1–10% of the population [2,4,5,18,19,30,32] (Figure 6). Usually, aRHA and aLHA are present separately in the population but in some cases, both variations are present together. According to a study, the combined variation of aRHA and aLHA is present in 0.27% of the study population [37]. In the Michel type II variation, replaced LHA mainly supplies segments II and III, and segment IV receives branches from the RHA. In some cases, the arterial supply of segment IV is coming from replaced LHA. In case of the accessory LHA, segments II and III have dual circulation coming from the main branch of LHA and accessory LHA [3,23,33]. In the condition of ligation of LHA during elective surgeries and trauma in case of aberrant LHA, either replaced or accessory, it is well managed or tolerated due to collateral circulation maintained by arterial branches of the right hemiliver and collaterals between the intrahepatic and phrenic arteries; hence, there are very low chances of the development of hepatic necrosis and other complications [34,35,38,39]. Knowledge of the extrahepatic and intrahepatic branching pattern of variant LHA is important in gastric and liver surgeries for the proper maintenance of the blood supply, especially the left lobe of the liver.

## 4. Detection of Hepatic Artery Variations and Their Classifications

For the detection of accurate anatomy of the hepatic artery and its variations, MDCT and MRI are the most commonly used technique, especially for the detection of the aberrant artery. Axial CT scan has 96.3% sensitivity and 87% specificity and 88% accuracy for the detection of aberrant RHA [40,41]. By the use of MDCT, preoperative identification of the hepatic artery variations is helpful in surgical planning and its exact approach. Few studies have described the modified surgical approach of pancreatoduodenectomy in the presence of aRHA identified by hepatobiliary imaging by CT scan [42,43]. MDCT is a non-invasive technique as compared to routine angiography to visualize the liver, pancreas, biliary apparatus and associated tumors. The commencement of the MDCT scanner has greatly increased the productivity and usefulness of CT angiography in clinical practice [44]. Even though there are advanced CT imaging techniques, sometimes, arterial anomalies may be missed. Therefore, to prevent vessel injury, arterial pulsation may be felt before the dissection, i.e., at the lateral border of the hepatobiliary ligament in normal anatomy or behind the head of the pancreas or portal vein in case of aRHA and aberrant CHA.

Various classifications are given to explain the variations of the hepatic artery. Firstly, in 1966, Michel identified the hepatic artery variations and classified them into 10 different types [4]. Hiatt simplified and classified the hepatic artery variations into six different types [5]. The classification systems of Michel and Hiatt are most commonly accepted because of their simplicity. There are some flaws in these classifications as they do not explain the rare type of hepatic artery variations as the origin from the splenic and gastroduodenal arteries. Another drawback of these classifications is no explanation of the route of the hepatic artery which is a significant factor in hepatobiliary and pancreatic surgical planning. Some scientists explain the rare variations of the hepatic artery. Kobayashi et al. identified rare variations of the hepatic artery and classified hepatic artery variations into four major groups “Y”, “I-I”, “Y plus I”, and “I-I plus I” [45] (Table 2). A study has also explained some rare hepatic artery variations as LHA and RHA separately originated from the celiac trunk (gastroduodenal artery from RHA), LHA and RHA separately originated from the celiac trunk (gastroduodenal artery from LHA), accessory RHA separately originated from the celiac trunk, LGA originated from the gastroduodenal artery and RHA from SMA and LHA originated from the gastroduodenal artery [46]. Knowledge of rare variations of the hepatic artery is necessary for hepatobiliary surgeons to prevent intraoperative bleeding and complications during the surgery. In the literature, no single classification system covers all the aspects of hepatic artery variations.

## 5. Clinical Applications of the Hepatic Artery Variations

Precise knowledge of hepatic artery variations is helpful in the preoperative planning of surgeries such as liver tumor resection, liver transplantation, pancreatic tumor resection, pancreatoduodenectomy and biliary surgeries, which can reduce intraoperative bleeding complications and improve postoperative surgical outcomes.

### 5.1. Pancreatoduodenectomy

In the patients presenting with carcinoma of the head of the pancreas, the periampullary region and distal biliary tree surgical treatment is the modality of choice [47] (Figure 7). To successfully perform these complicated surgeries, one should thoroughly know the vascular anatomy of the hepatic artery. aRHA is expected to be seen in one in five patients experiencing pancreatic and biliary surgery. The identification and appropriate management of aRHA is a critical task in pancreatic surgery because any damage to the aberrant artery leads to complications such as liver ischemia, necrosis and biliary anastomosis leak [48,49]. For pancreatoduodenectomy in case of gastric carcinoma, replaced LHA arising from the left gastric artery should be preserved; otherwise, liver necrosis occurs as a postoperative complication [50]. To some extent, portal venous circulation of the liver parenchyma compensates the liver ischemia. Other factors such as extrahepatic collateral circulation and interlobular communicating artery also have protective effects and help to prevent liver ischemia [51]. Various radiological studies showed the collateral circulation of the liver in case of blockage of the main artery; supply is maintained by the inferior phrenic artery, superior and inferior epigastric artery and some gastric collaterals [4,52,53]. According to the available literature, in more than 20% of cases, liver ischemia and necrosis are seen as postoperative complications [50,52]. During pancreatoduodenectomy, it is important to maintain the adequate arterial supply of the common bile duct, as blockage of the arterial supply causes the biliary fistula. The main arterial supply of the common bile duct comes from the retroduodenal artery branch of the gastroduodenal artery and intra- and peribiliary vessels present in porta hepatic. In pancreatoduodenectomy surgery, intra- and peribiliary vessels as well as the gastroduodenal artery are routinely sacrificed and removed thereafter; the bile duct is completely in need of RHA for its arterial supply [50,54]. Cancer of the head of the pancreas is often associated with biliary obstruction leading to obstructive jaundice. High concentrations of bile acid in the liver cell predispose them to hypoxia and mitigate the effect of liver ischemia [55]. Normally, pancreatoduodenectomy is associated with 13 to 35% of morbidity due to pancreatic leak. The presence of hepatic artery variations is itself a challenging task for surgeons to maintain appropriate vital arterial supply during the surgical procedure. There is always a matter of debate among scientists that hepatic artery variations during pancreatic surgery increase morbidity and overall survival.

### 5.2. Management of Aberrant Hepatic Artery during the Surgical Procedure

Proper anatomical knowledge of the aberrant hepatic artery is crucial in complicated upper abdominal surgical procedures. aRHA arising from the SMA is the most common variation of the hepatic artery; it passes beneath the pancreatic head and bile duct, and lateral to the portal vein. A less frequent variation is CHA originating from the SMA or aorta beneath the pancreatic head. Preoperative identification of the aberrant hepatic artery with the help of the MDCT and planning the management is the ideal approach [56]. Sometimes, unrehearsed surgical decisions may be taken by the surgeons due to specific tumor involvement and arterial variations [57]. Four available surgical options for managing aberrant vessels include sacrifice, preoperative embolization, dissection and preservation, and transaction and reconstruction [49,58,59,60,61,62]. For the small aberrant hepatic artery, sacrifice is an ideal approach because it does not affect the clinically significant outcome of surgery [58]. The aberrant hepatic artery with a large caliber has more tumor involvement; resection is necessary for adequate surgical outcome. Preoperative embolization is performed for the maintenance of collateral circulation to the right lobe and bile duct [61]. Dissection and preservation are the ideal approaches for all aberrant arterial variations but are not conceivable in all cases. Transaction and reconstruction are performed to construct primary anastomosis and implantation at the arterial site. If the small artery is involved, primary anastomosis is sufficient, but when a larger artery is involved, the primary artery is resected and implantation is performed [57]. In the case of pancreatic carcinoma, when the aberrant hepatic artery passes through the pancreatic head or uncinate process, then preoperative identification of the aberrant artery by the use of MDCT scan helps to modify the approach of pancreatoduodenectomy. In the condition of cancer of the head of the pancreas, if the aberrant artery is present and passing through the head, then transaction of the artery is inevitable. To reconstruct the small gap in the artery, primary anastomosis is performed, and if the segment is long, reconstruction with reimplantation is performed [57,63]. For reimplantation gastroduodenal artery, LGA and middle colic artery are commonly used [62,64,65]. In patients with bile duct carcinoma in the presence of aRHA, the aberrant artery is dissected through the ventromedial site of the hepatoduodenal ligament by preserving the CHA, LHA, proper hepatic artery and portal vein in pancreatoduodenectomy [66]. Some surgeons performed retropancreatic dissection to visualize aRHA and dissect it safely [43]. Normally, aRHA is exposed by dissecting the celiac lymph node. During the exposure of the aberrant artery in Kocher’s maneuver and retropancreatic dissection, precaution must be taken because excessive traction of the head of the pancreas may cause thrombosis of aRHA [57].

Extrahepatic circulation of the liver acts as collateral circulation in case of blockage of major vessels. These extrahepatic collaterals are mainly supplied by the epigastric arteries, inferior phrenic arteries and small gastric arteries [52,53,67,68]. In the aberrant course of the hepatic artery, proper identification of the intrahepatic branching pattern is important to know segmental blood supply in complicated gastric, bariatric, and liver surgeries. The liver shows well-developed intrahepatic and extrahepatic collateral pathways in cases of the blocked main hepatic artery [69]. Due to the presence of these collateral pathways, liver parenchyma can well sustain the main hepatic artery embolization [70]. Preoperative embolization of the hepatic artery is performed for the development of the collateral pathways in case of an aberrant hepatic artery for minimizing ischemia-related complications. Some researchers have explained that there is no need for embolization if the liver has sufficient collateral pathways. The preoperative embolization of CHA in the case of pancreatic head carcinoma with aberrant CHA is safe to prevent ischemia-related complications before the resection of CHA [61,71]. According to some literature, there is no difference in postoperative bleeding complications in patients with aberrant hepatic arteries versus normal hepatic arteries [72,73,74]. Trang et al. and Kim et al. both described robotic pancreatoduodenectomy in the presence of aberrant hepatic artery and compared it with open pancreatoduodenectomy in relation to safety and postoperative bleeding complications [75,76].

### 5.3. Liver Transplantation

For successful liver transplantation, the maintenance of proper blood supply is necessary. Hepatic artery variations of liver donors are associated with complex arterial construction and might be associated with arterial bleeding complications in orthotopic liver transplantation (OLT). At the time of surgery, most of the anatomical variations remain undetected because of their negligible effect on the surgical outcome. To avoid injury to accessory vessels supplying the liver, careful dissection is performed to palpate the hepatic artery. Hepatic artery variations also have an effect on early bile duct stenosis and necrosis within the limited range because the major bile duct is also supplied by the anastomosis between the RHA and LHA [77,78,79]. The major complication related to OLT is hepatic artery thrombosis (HAT). Variations of the major hepatic artery such as CHA, RHA and LHA are associated with the large reconstruction, which is the major factor for the development of HAT. At the same time, anatomical knowledge of the intrahepatic branching pattern of variant hepatic arteries to particular segments of the liver is of utmost importance to minimize the ischemia-related complications, especially for the segments that have peculiarities in their blood supply [33]. The rejection of the graft due to HAT is seen in 3–9% of cases of OLT [80]. The liver has a sufficient amount of collateral circulation to maintain blood flow during HAT [81]. According to the available literature, one-third of cases of HAT are asymptomatic, one-third of cases are asymptomatic in the initial stage but later develop biliary tract ischemia, and one-third of cases develop very serious life-threatening complications such as parenchymal ischemia and necrosis, and a lack of prompt action may lead to death [82,83]. Anomalies in the anatomy of hepatic arteries are a major risk factor for the development of HAT. Aberrant hepatic arteries always require more reconstruction and lead to more chances of HAT associated with this anatomical variation. In the case of aRHA, which is replaced and originates from SMA, more attention in OLT is required, as reconstruction anastomosis between SMA and celiac trunk produce more twisting and backflow, making them more prone for the development of HAT [84]. In the condition of accessory RHA along with normal RHA, the section of accessory branches has no serious complication such as hepatic necrosis because arterial supply is maintained by the main normal trunk of RHA. The aberrant left hepatic artery is also named “Hyrtl’s artery”, which is “replaced” when it does not originate from the hepatic proper artery and is the only supply to that part of the liver, while an accessory left hepatic artery coexists with a normal left hepatic artery [39]. During a liver procurement for liver transplantation, the section of a small accessory left hepatic artery has no severe consequences on the liver graft. The sacrifice of a replaced hepatic artery can have serious consequences for the transplanted liver graft and should be reconstructed during bench surgery. The hepatic artery variations which require more reconstruction anastomosis are more prone to the development of HAT as compared to less reconstruction anastomosis in the donor and recipient both in OLT [85,86].

Knowledge of hepatic artery anatomy and its variations for both the donor and recipient is essential for liver transplantation surgery. In cadaveric liver transplantation, the anatomical knowledge of a hepatic artery is important to avoid arterial injury and to improve the plan of reconstruction during the surgery [87]. The incidence of hepatic artery variations ranges from 10 to 30% in hepatic grafts [87,88]. The most commonly seen hepatic artery variations are RHA originating from the SMA followed by LHA from the LGA. In the case of graft, aberrant RHA from SMA need extensive reconstruction during the surgical procedure for preventing ischemia; most commonly, RHA is reconstructed with the gastroduodenal artery to maintain the short length of the artery to minimize HAT, in some cases with the splenic artery [87,89,90,91]. In the case of replaced LHA, all cases do not require reconstruction, but some cases which are prone to the development of ischemia or biliary strictures require reconstruction [90]. If replaced LHA is resected during the liver harvesting, firstly it is checked for no back bleeding from the root of resected replaced LHA and then anastomosed with the LGA. In the condition of failure to establish proper anastomosis, ischemia of segments II and III occurs [89]. Anatomical variations of the hepatic artery play a key role in orthotopic and split liver transplants. Recipient arterial anomalies do not affect the long-term and short-term output, but donor anomalies affect the outcome. In particular, RHA arising from the SMA needs more attention as it is associated with long cold ischemic time, more blood cell requirements and proper reconstruction.

At the time of liver transplantation surgery, the endothelium of the graft becomes activated due to reconstruction and ischemia. Due to the activation of the endothelium, certain chemical factors are released which attract the platelets and cause platelets aggregation. This leads to the formation of large thrombus, and it also triggers procoagulant factors, which further activate the platelets and initiate the coagulation process [92]. Some studies explain the role of anticoagulants such as aspirin in preventing postoperative complications such as thrombosis and clot formation [93,94].

The survival of patients in liver transplantation depends upon the early identification of HAT and in case of life-threatening ischemia and necrosis of the liver, prompt management and repeat OLT is required [95,96,97]. According to Dala Riva et al., the presence of hepatic arterial variations increases the risk of thrombosis seven-folds, and the combination of arterial variations with the reconstruction of arterial anastomosis increases the risk of thrombosis by eighteen folds [80].

## 6. Hepatic Artery in Interventional Radiology

The liver is the most commonly injured organ during blunt abdominal trauma of the abdomen region followed by the spleen [98]. For the management of liver trauma, surgical options are limited and associated with higher unsuccessful rates and mortality. With new advances in imaging with the MDCT, the non-operative management of blunt liver trauma is considered the treatment of choice in hemodynamically stable patients [99,100]. Transarterial angioembolization is considered the gold-standard non-operative technique in the management of traumatic liver injury with a high success rate [101]. CT angiography and angioembolization are key components of the non-operative management of blunt liver injuries. Although angioembolization is a very effective modality, it has some major limitations such as necrosis of the liver and biliary tract, abscess and cholecystitis [98]. Due to the dual blood supply of the liver, extensive angioembolization is well tolerated. In case of injury to the proper hepatic artery, it is firstly repaired, and if bleeding does not stop, then the vessel is ligated. In the ligation of RHA and CHA, cholecystectomy should be performed due to the risk of necrosis. Injury to the portal vein is crucial and should be repaired properly, since ligation of the vein leads to massive liver necrosis and ischemia. In the condition where portal vein ligation is necessary, firstly, adequate arterial circulation of liver parenchyma should be ensured. In unrepairable intraparenchymal bleed, resection of the affected segment can be performed, and in case of extrahepatic bleeding, liver packing can be used to stop the bleeding if direct repair is not achieved [102].

Tumors of hepatocellular carcinoma (HCC) mostly receive their blood supply from the hepatic artery. Hepatic artery chemoembolization is used as a treatment modality for HCC tumors by reducing their blood supply to induce hypoxic tumor cell death [103]. Chemoembolization for the treatment of hepatocellular carcinoma is also termed as trans-arterial chemoembolization (TACE). Liver function must be monitored adequately in patients undergoing TACE because it increases the risk of acute liver failure. The risk is significantly increased in patients with Child–Pugh B cirrhosis [104]. Chemoembolization of the hepatic artery may compromise the hepatic parenchymal perfusion and result in an acute worsening of liver function.

In both angioembolization and chemoembolization, proper hepatic artery embolization would be prone to develop whole liver ischemia. In most of cases, adequate portal venous flow maintains the viability of the liver parenchyma in conditions of compromised hepatic arterial blood flow. Portal venous flow has low pressure, and in combination with injury to the portal vein, it may well compromise and cause liver ischemia. For preventing hepatic necrosis, multiple sites of embolization of hepatic artery branches may be marker of injury [105]. Superselective arterial catheterization for more focused drug delivery is the golden rule to prevent liver necrosis [106].

## 7. Comparison of Different Imaging Methods in Detecting Arterial Anatomy

The most commonly used imaging methods to detect vascular variations and diseases are computed tomographic angiography (CTA), magnetic resonance angiography (MRA) and digital subtraction angiography (DSA). All the stated methods have their respective advantages and disadvantages in the detection of arterial anatomy. DSA is considered the gold standard for the visualization of small arteries and their pathologies such as aneurysms and arteriovenous malformations but is less commonly used because of its invasive nature and high diagnostic cost [107]. Three-dimensional (3D) DSA is a more recently used method and has various advantages over 2D DSA [108]. Lately, the combination of 2D and 3D DSA is commonly used to identify vascular pathologies [109].

CTA and MRA imaging techniques are more sensitive and specific in the diagnosis of vascular pathologies. CTA is non-invasive and cost-effective in the detection and localization of vascular pathologies in comparison to DSA [110]. In CTA, image processing is fast because of less imaging time, which saves treatment time for patients. Another advantage of CTA is the minimization of radial artifacts. CTA can also detect a 3D view of the blood vessels along with their clinical condition. For the diagnosis of vascular pathologies of small blood vessels, CTA is preferred due to its high detection rate [108]. CTA is widely used in clinical practice but has disadvantages such as exposure to X-ray radiation along with allergic reactions due to the injection of contrast agents [111].

MRA is commonly used for the diagnosis of vascular diseases due to its higher specificity and non-invasive nature. In the MRA technique, there is no need of injecting contrast agents, and it is contraindicated in patients having metal in the body. According to Feng et al., the main disadvantage of MRA is the early saturation of small peripheral blood vessels affecting the final observation [108]. In comparison to the CTA and DSA, MRA has low sensitivity and accuracy in the detection of vascular pathologies [112].

## 8. Latest Diagnostic Imaging Modalities to Detect Liver Vascularity

Advanced imaging modalities play an important role in the detection of accurate liver vascularity along with its variability. With the availability of modern imaging modalities, exact microvascular circulation with the segmental perfusion of the liver can be very well detected. These latest imaging modalities include ultrasound, contrast-enhanced ultrasound, MDCT, perfusion CT scan, MRI, contrast-enhanced MRI, positron emission tomography (PET), digital subtraction angiography (DSA), etc.

### 8.1. Ultrasound

Doppler liver ultrasound is a cost-effective and non-invasive method for the detection of liver vascularity and its variation. By the use of liver ultrasound, vascular pattern of the hepatic artery, hepatic vein and portal vein can be detected [113,114]. Ultrasound includes the spectrum of techniques such as B-mode, color, and power Doppler techniques (plus wave Doppler and non-Doppler flow visualization) and recently, contrast-enhanced ultrasound. Nowadays, the latest non-Doppler techniques such as superb microvascular imaging or b-flow/high-definition color are used to overcome the limitations of Doppler [115].

Contrast-enhanced ultrasound is the combination of the contrast agent with the modern imaging techniques such as pulse-inversion technique Due to this combination, the sensitivity and specificity of ultrasound is dramatically increased. In contrast-enhanced ultrasound, an arterial phase, portal-venous phase, and late arterial phase can be produced [116]. Contrast-enhanced ultrasound is recommended in cases of renal failure or indecisive findings in MDCT and MRI.

### 8.2. CT Scan

MDCT plays a very important role in detecting the intrahepatic and extrahepatic vascular pattern of the liver. The most advanced CT scans use wide detector arrays of more than eight row detectors for imaging [117]. A CT scan consists of non-contrast and contrast-enhanced images. For the detection of fat deposition and its differentiation from blood products, a non-contrast CT scan is commonly used. MDCT consists of the arterial phase, portal phase, venous phase, and delayed phase. In the arterial phase of the CT scan, a full arterial enhancement of liver parenchyma is very well seen. In the portal-venous phase, maximum enhancement of the portal vein with moderate enhancement of the hepatic vein is obtained [118]. **A** CT scan is well tolerated and less prone to motion artifacts as compared with MRI. Radiation exposure is the main disadvantage of CT scans. The sensitivity and specificity of MDCT are lower than those of MRI.

The perfusion CT scan is a newly developed technique used to detect the microcirculation of liver parenchyma as well as the morphology and characteristics of focal liver lesions. The hypervascular liver lesions are very well detected by MDCT, but the small hypovascular lesion can be misinterpreted. Perfusion CT scan can accurately detect the small hypovascular lesion of the liver [118,119].

### 8.3. MRI

MRI of the liver provides more detailed information about tissue characteristics of the liver tissue along with detailed information about its vascularity. In comparison to MDCT, the sensitivity and specificity of MRI are more for the detection of focal and diffuse liver lesions. MRI is a superior technique to detect fatty changes in the liver from the CT scan [120]. With the introduction of hepatospecific contrast agents, a new path of liver imaging has been introduced. MRI is a superior imaging modality with a higher resolution to access functional and morphological characteristics of the liver [118]. MRI not only diagnoses the lesion but also accesses the prognostic parameters, which has a direct effect on the clinical outcome.

MR perfusion also provides detailed information about tissue microcirculation. In the liver, dynamic contrast-enhancement MRI is the most frequently used approach which requires gadolinium contrast administration with the acquisition of a single time curve [121]. The main challenges for liver MR perfusion imaging are the dual blood supply of the liver, sinusoids, and respiratory movements [122].

### 8.4. Positron Emission Tomography with Computed Tomography (PET/CT)

Positron emission tomography with [18F] fluoro-2-deoxy-d-glucose (FDG) is a very useful modality for the diagnosis of various focal lesions. The accuracy of PET dramatically increases in combination with the computed tomography technique. PET/CT is commonly used for the detection of tumors, to perform its staging and for the prediction of treatment response [123].

### 8.5. Digital Subtraction Angiography (DSA)

Digital subtraction angiography (DSA) is the preferred method for visualizing the vascularity of the liver. With the use of DSA, the hepatic artery branching pattern along with its intrahepatic branching is very well studied. DSA can visualize the small vessels very accurately with their pathology. Time-resolved three-dimensional digital subtraction angiography (4D-DSA) is the most recently developed angiographic method providing 3D quantitative information about blood flow [124]. Four-dimensional (4D)-DSA has limited access in the thorax and abdominal region because of respiratory motion and patient susceptibility [125]. The major limitation of DSA is its invasive nature.

## 9. Conclusions

The review focuses on hepatic artery variations and their classification on the basis of Michel and Hiatt’s classification system mainly, and it also describes the embryological basis of these variations. We are also focusing on some rare variations explained by Kobayashi in their classification, which were not included in Michel and Hiatt’s classification. The study is more focused on aberrant hepatic arterial variations and their exact anatomical location by the use of advanced imaging techniques such as MDCT. The review also throws light upon various advantages and disadvantages of different imaging techniques such as DSA, MRA and CTA, and it also discusses the latest diagnostic and therapeutic imaging modalities and intervention radiology. The review draws attention toward exact anatomical knowledge of hepatic artery variations and their clinical importance to minimized intraoperative difficulties and the improvement of postoperative surgical outcomes in complicated surgeries such as liver transplantation, pancreatoduodenectomy, biliary reconstruction and gastric surgeries.

## Figures and Tables

**Figure 1 diagnostics-13-01233-f001:**
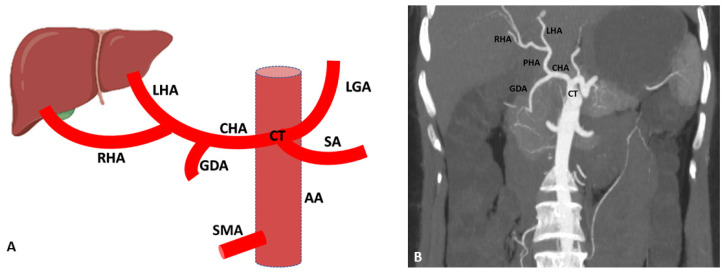
(**A**) Schematic representation of regular branching of the hepatic artery. (**B**) Coronal maximum intensity projection (MIP) (64-row scanner CT angiography) image of the abdominal region showing the regular anatomical branching pattern of the hepatic artery. Slice thickness: 1.2 mm. AA: Abdominal aorta; CT: Celiac trunk; SA: Splenic artery; LGA: Left gastric artery; CHA: Common hepatic artery; PHA: Proper hepatic artery; RHA: Right hepatic artery; LHA: Left hepatic artery; GDA: Gastroduodenal artery; SMA: Superior mesenteric artery.

**Figure 2 diagnostics-13-01233-f002:**
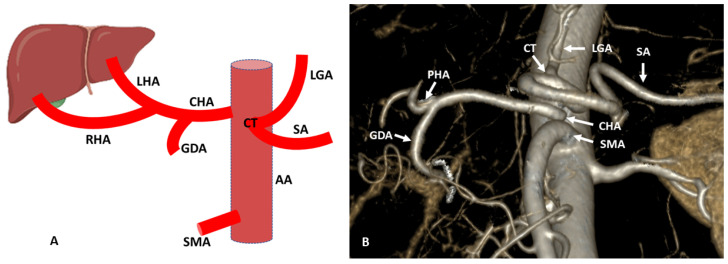
(**A**) Schematic representation of the origin of CHA from the aorta. (**B**) Three-dimensional (3D) volume render (3D-VR) image (64-row scanner CT angiography) of the abdominal region showing the aberrant (replaced) CHA originating from the aorta. AA: Abdominal aorta; CT: Celiac trunk; SA: Splenic artery; LGA: Left gastric artery; CHA: Common hepatic artery; PHA: Proper hepatic artery; RHA: Right hepatic artery; LHA: Left hepatic artery; GDA: Gastroduodenal artery; SMA: Superior mesenteric artery.

**Figure 3 diagnostics-13-01233-f003:**
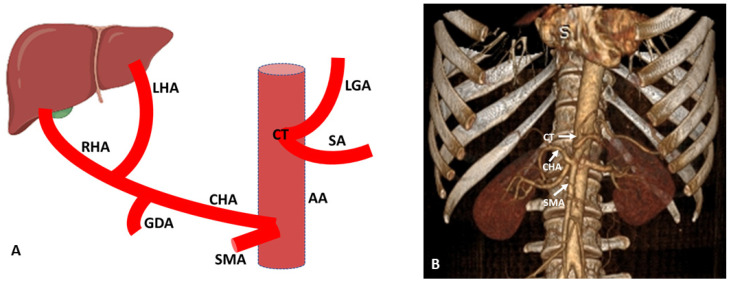
(**A**) Schematic representation of aberrant (replaced) CHA originating from SMA. (**B**) Three-dimensional (3D) volume render (3D-VR) image (64-row scanner CT angiography) of the abdominal region showing the aberrant (replaced) CHA originating from SMA. AA: Abdominal aorta; CT: Celiac trunk; SA: Splenic artery; LGA: Left gastric artery; CHA: Common hepatic artery; RHA: Right hepatic artery; LHA: Left hepatic artery; GDA: Gastroduodenal artery; SMA: Superior mesenteric artery.

**Figure 4 diagnostics-13-01233-f004:**
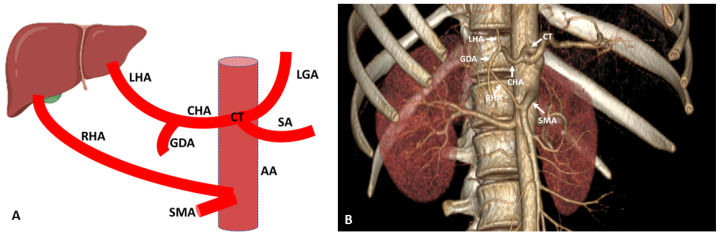
(**A**) Schematic representation of aberrant (replace) RHA originated from SMA. (**B**) Three-dimensional (3D)-VR image (64 row CT-angiography) of the abdominal region showing aberrant (replaced) RHA originated from SMA. AA: Abdominal aorta; CT: Celiac trunk; SA: Splenic artery; LGA: Left gastric artery; CHA: Common hepatic artery; RHA: Right hepatic artery; LHA: Left hepatic artery; GDA: Gastroduodenal artery; SMA: Superior mesenteric artery.

**Figure 5 diagnostics-13-01233-f005:**
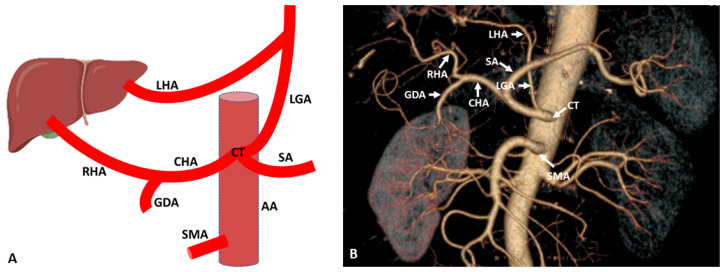
(**A**) Schematic representation of aberrant (replaced) LHA originated from LGA. (**B**) Three-dimensional (3D)-VR image (64 row CT-angiography) of the abdominal region showing aberrant (replaced) LHA originated from LGA. AA: Abdominal aorta; CT: Celiac trunk; SA: Splenic artery; LGA: Left gastric artery; CHA: Common hepatic artery; RHA: Right hepatic artery; LHA: Left hepatic artery; GDA: Gastroduodenal artery; SMA: Superior mesenteric artery.

**Figure 6 diagnostics-13-01233-f006:**
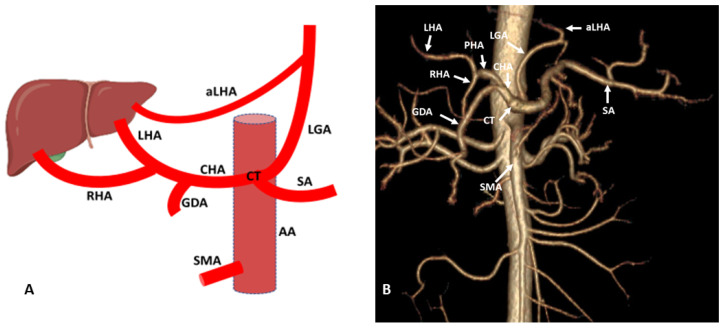
(**A**) Schematic representation of accessory LHA (aLHA) originated from LGA. (**B**) Three-dimensional (3D) volume render (3D-VR) image (64-row scanner CT angiography) of the abdominal region showing the accessory LHA (aLHA) originated from LGA. AA: Abdominal aorta; CT: Celiac trunk; SA: Splenic artery; LGA: Left gastric artery; CHA: Common hepatic artery; PHA: Proper hepatic artery; RHA: Right hepatic artery; LHA: Left hepatic artery; GDA: Gastroduodenal artery; SMA: Superior mesenteric artery.

**Figure 7 diagnostics-13-01233-f007:**
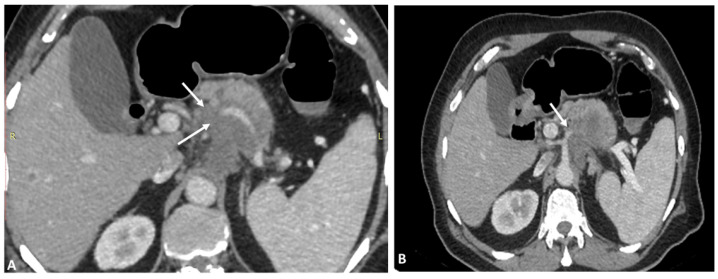
(**A**) Axial phase maximum intensity projection (MIP) (64-row scanner CT angiography) image of the abdominal region showing tumor infiltrating pancreatic head indicated by the arrow. (**B**) Axial phase maximum intensity projection (MIP) (64-row scanner CT angiography) image of the abdominal region showing tumor invading the artery indicated by the arrow.

**Table 1 diagnostics-13-01233-t001:** Showing the anatomical variations of the hepatic artery based on the classification of Michel and Hiatt.

Anatomical Variation of Hepatic Artery	Michel Classification	Hiatt Classification
Normal anatomy	Type I	Type I
LHA branch LGA	Type II	Type II
RHA branch SMA	Type III	Type III
Type I and II association	Type IV	Type IV
LHA accessory LGA	Type V	Type II
RHA accessory SMA	Type VI	Type III
LHA accessory LGA + RHA accessory SMA	Type VII	Type IV
LHA accessory LGA+ RHA branch SMA	Type VIII	Type IV
CHA branch SMA	Type IX	Type V
RHA and LHA branch LGA	Type X	------
CHA aorta branch	------	Type VI

**Table 2 diagnostics-13-01233-t002:** Showing the anatomical variations of the hepatic artery based on the Kobayashi classification system.

S. No.	Category	Subcategory
1.	“Y”	(i) “Y”; CHA (normal anatomy)
		(ii) “Y”; CHA-CMA (celiomesentric trunk type)
		(iii) “Y”; CHA-SMA (CHA from SMA)
		(iv) “Y”; CHA-Ao (CHA from aorta)
2.	“Y plus I”	(i) “I, Y”; SMA, CHA (accessory RHA from SMA)
		(ii) “I, Y”; Ao, CHA (accessory RHA from aorta)
		(iii) “I, Y”; GDA, CHA (accessory RHA from GDA)
		(iv) “I, Y”; CHA, LGA (accessory LHA from LGA)
		(v) “I, Y”, I”; SMA, CHA, LGA (accessory RHA from SMA and accessory LHA from LGA)
3.	“I-I”	(i) “I-I”; SMA, CHA (RHA from SMA)
		(ii) “I-I”; GDA, CHA (RHA from GDA)
		(iii) “I-I”; CHA, LGA (LHA from LGA)
		(iv) “I-I”; CHA, GDA (LHA from GDA)
		(v) “I-I”; CHA, LGA-Ao (LHA from LGA and LGA from aorta)
		(vi) “I-I”; CHA, Ao (RHA & LHA separately from aorta and GDA from RHA)
		(vii) “I-I”; Ao, CHA (RHA & LHA separately from aorta and GDA from LHA)
		(viii) “I-I”; SMA, GDA (LHA from GDA and RHA from SMA)
		(ix) “I-I”; SMA, Ao (LHA from aorta and RHA from SMA)
		(x) “I-I”; SMA, LGA (LHA from LGA and RHA from SMA)
4.	“I-I plus I”	(i) “I-I, I”; SMA, CHA, LGA (accessory LHA from LGA and RHA from SMA)
		(ii) “I, I-I”; SMA, CHA, LGA (LHA from LGA and accessory RHA from SMA)

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
