# Peer review of "Importance of Anatomical Variation of the Hepatic Artery for Complicated Liver and Pancreatic Surgeries: A Review Emphasizing Origin and Branching"

_diagnostics, 2023, doi:10.3390/diagnostics13071233_

Round 1

Reviewer 1 Report

The review focuses on hepatic artery variations and their classification based on Michel and Hiatt’s classification system mainly and also describes the embryological basis of these variations. This is a very practical review, which has a good guiding role for clinical work.

 However, there are several additional suggestions:

 1. It is suggested that each arterial variation should be equipped with a CT image, marked with the arterial name, and attached with a schematic diagram to show the arterial variation so that this article can become a leader in similar articles and increase the number of citations. 

2. In the Clinical applications of the hepatic artery variations section, the author lists three applications and suggests that imaging images can also be supplemented separately, such as showing the boundary between pancreatic head cancer and artery. If there is cancer invading the artery, the arrow can be used to point out the area of arterial invasion, which can better help readers understand the importance of arterial variation. 

3. Can you add a paragraph to explain and compare the advantages and disadvantages of different imaging methods in displaying arterial variation, such as the role of CTA, MRA, and DSA in arterial variation?

Reviewer 2 Report

In the manuscript entitled "Importance of anatomical variation of the hepatic artery for 2 complicated liver and pancreatic surgeries: A review empha- 3 sising on origin and branching” by Kapil Kumar Malviya et al., the authors discusses variations in the anatomy of the hepatic artery from its origin, to branching by the use of advanced imaging techniques and its effect on the liver, pancreatic, biliary and gastric surgeries.

Major concern:

The mentioned method has been utilized in different articles. So, the author should mention and clarify the difference between them and the others.

Kindly, try to write the article in a novel way to be accepted for publication.

Minor concerns:

The reference list needs to be updated with the last three years, particularly before 2010.

Reviewer 3 Report

The paper is a deep revision of the anatomical variants of hepatic artery by a surgical and clinical point of view.

It is useful between young and expert surgeons in order to improve their practice.

Round 2

Reviewer 2 Report

Authors did necessary change.

Author Response

Dr. Kapil Kumar Malviya                                                      Assistant Professor

                                                                                                Department of Anatomy

Institute of Medical Sciences

                                                                                                Banaras Hindu University

                                                                                                Varanasi-221005

Dt : 05-03-2023

To,

Dr. Anna Żurada

Guest Editor

Special Issue "Clinical Anatomy: Advances and Applications in Diagnostics"

Diagnostics

MDPI

Subject: Submission of second revision of the invited review article titled “Importance of anatomical variation of the hepatic artery for complicated liver and pancreatic surgeries: A review emphasising on origin and branching” in special issue "Clinical Anatomy: Advances and Applications in Diagnostics " of Diagnostics, MDPI.

Dear Dr. Żurada,

Please find attached herewith second revision of the invited review article entitled “Importance of anatomical variation of the hepatic artery for complicated liver and pancreatic surgeries: A review emphasising on origin and branching” by Kapil Kumar Malviya, Ashish Verma for publication in special issue "Clinical Anatomy: Advances and Applications in Diagnostics" of Diagnostics after incorporation of the reviewer’s and editor´s suggestions and comments. We are thankful to academic editor and reviewer for evaluating our review article and appreciating our work and providing useful comments and suggestions. These suggestions and comments have improved the clarity, presentation and impact of our review. All the revisions are highlighted in green and indicated through review mode in the revised review. All the revisions suggested by the editor are indicated by line numbers in “response to academic editor and reviewer comments”.

Knowledge of anatomical variations of the hepatic artery from its origin to intrahepatic segmentation is of utmost importance for planning upper abdominal surgeries. The origin and branching pattern of the hepatic artery was thoroughly described by the Michels and Hiatts classification and some rare variations of the hepatic artery were classified by Kobayashi and Koops. By the use of the multidetector computed tomography (MDCT) technique, the branching pattern of the hepatic artery can be visualized quite accurately. The origin and course of the aberrant hepatic arteries are very much crucial in surgical planning of carcinoma of the head of the pancreas and hepatobiliary surgeries. This review discusses variations in the anatomy of the hepatic artery from its origin, to branching by the use of advanced imaging techniques and its effect on the liver, pancreatic, biliary and gastric surgeries.

The revised review article contains two tables and seven figures.

We look forward to our publication in the special issue "Clinical Anatomy: Advances and Applications in Diagnostics" of Diagnostics.

Thanking you

With best regards

Dr. Kapil Kumar Malviya

Manuscript ID: Diagnostics- 2223377

Importance of anatomical variation of the hepatic artery for complicated liver and pancreatic surgeries: A review emphasising on origin and branching

Kapil Kumar Malviya a*, Ashish Verma b*

a Department of Anatomy, Institute of Medical Science, Banaras Hindu University, Varanasi, U.P., India

b Department of Radiodiagnosis, Institute of Medical Science, Banaras Hindu University, Varanasi, U.P., India

* Correspondence: Corresponding authors: -Dr. Ashish Verma, Professor & Head, Department of Radiodiagnosis and Imaging, Institute of Medical Sciences Banaras Hindu University, E-mail address: [email protected]

-Dr. Kapil Kumar Malviya, Assistant Professor, Department of Anatomy Institute of Medical Sciences Banaras Hindu University, Tel.: +91 8052631840 (mobile). E-mail addresses: [email protected] , [email protected]

RESPONSE TO REVIEWERS COMMENTS

Reviewer´s comments:

Reviewer #2:

Comments and Suggestions for Authors

Authors did necessary change.

Response: Authors thank the reviewer for appreciating and encouraging words on our review. We thank the reviewer for providing valuable time in evaluating our review.
